# Detection and Risk Analysis with Lane-Changing Decision Algorithms for Autonomous Vehicles

**DOI:** 10.3390/s22218148

**Published:** 2022-10-24

**Authors:** Amin Mechernene, Vincent Judalet, Ahmed Chaibet, Moussa Boukhnifer

**Affiliations:** 1ESTACA Engineering School, 12 Rue Paul Delouvrier, 78180 Montigny-le-Bretonneux, France; 2DRIVE, Université de Bourgogne, 49 rue Mademoiselle Bourgeois, BP 31, CEDEX, 58027 Nevers, France; 3Université de Lorraine, LCOMS, F-57000 Metz, France

**Keywords:** risk assessment, decision making, autonomous driving, lane-changing maneuver, decision tree, random forest, artificial neural network, driving simulator

## Abstract

Despite the great technological advances in ADAS, autonomous driving still faces many challenges. Among them is improving decision-making algorithms so that vehicles can make the right decision inspired by human driving. Not only must these decisions ensure the safety of the car occupants and the other road users, but they have to be understandable by them. This article focuses on decision-making algorithms for autonomous vehicles, specifically for lane changing on highways and sub-urban roads. The challenge to overcome is to develop a decision-making algorithm that combines fidelity to human behavior and that is based on machine learning, with a global structure that allows understanding the behavior of the algorithm and that is not opaque such as black box algorithms. To this end, a three-step decision-making method was developed: trajectory prediction of the surrounding vehicles, risk and gain computation associated with the maneuver and based on the predicted trajectories, and finally decision making. For the decision making, three algorithms: decision tree, random forest, and artificial neural network are proposed and compared based on a naturalistic driving database and a driving simulator.

## 1. Introduction

An autonomous vehicle (driver-less) is able to move without human intervention and interact and deal with its surroundings, such as pedestrians and other users road. To achieve this task, several technological bricks are needed. Sensors are required, such as radars, lidars, and cameras, combined with computing hardware and software to build a numerical map of the surrounding environment and make decisions and actions according to user requests and manufacturer settings. The concept of autonomous car covers, depending on the context, is a fully autonomous vehicle or a semi-autonomous vehicle with a variety of driving assistance systems. The taxonomy of driving automation is defined by the Society of Automotive Engineers (SAE) as follows [1]:Level 0: No Driving Automation.Level 1: Driver Assistance.Level 2: Partial Driving Automation.Level 3: Conditional Driving Automation.Level 4: High Driving Automation.Level 5: Full Driving Automation.

Nevertheless, fully autonomous vehicles are not widely available [2], and this despite the great technological advances in this field during the last ten years. There are still many challenges to be met in order to achieve a completely autonomous vehicle, which can operate whatever the conditions and ensure total safety, or at least safety superior to human drivers. Among the cons to overcome are:Improving the perception of the environment to provide a correct interpretation of the information returned by the sensors, whatever the lighting and weather conditions.Enhance decision-making algorithms so that vehicles can make the right decision, even if they encounter situations not foreseen during development but also so that they behave like humans because, during their travel, autonomous vehicles will have to share the road with non-autonomous vehicles, and it is important that other drivers can understand the actions of the autonomous vehicle.

In this paper, we are interested in decision-making algorithms for autonomous vehicles, specifically for lane changing on highways and arterial roads. The main objective is to develop a method to allow the vehicle to change lanes independently, ensuring the safety of passengers and other road users [3], lowering travel time. The main target is to reproduce human behavior, for two main reasons:Ensure passenger acceptance of the algorithm [4]: indeed, the vehicle must have a behavior similar to a human in terms of risk taking because if it is too careful or takes too much risk, the passengers may have a tendency to reject autonomous driving and regain control of the vehicle.To be understandable by other non-autonomous vehicles and pedestrians [5]: on the road, humans adopt behaviors to communicate with other road users non-verbally, such as slowing down to let a vehicle change lanes or pulling to the left of the current lane to signify an intention to change lanes. As long as autonomous vehicles will have to share the road with non-autonomous ones, it is important that they can understand these behaviors and behave the same.

There are already several methods for decision making, some based on a set of rules [6,7], and others based on artificial intelligence algorithms [8,9]. The existing methods aim at developing an optimal driving style while optimizing different criteria, such as the risk of collision or the occupant comfort, while scrupulously fulfilling the traffic regulation rules.

The problem of such a paradigm is that the driving style of the resulting automated pilot will necessarily differ from a human driver, which generally adopts a more intuitive and sometime less optimal driving style. In many situations, human drivers have to anticipate the behavior of other vehicles, for instance while trying an insertion maneuver in dense traffic, and an implicit cooperation between the drivers is sometimes necessary (a vehicle adapts its speed to enable the insertion of another vehicle). The introduction of automated vehicles with a “non-human” driving style (even through optimal) could perturb other human drivers, thus increasing the risk of accident for human road users.

Moreover, if an automated vehicle does not drive like a human, its behavior could be difficult to understand for the occupants, inducing frustration if the automated vehicle is too careful and not able to overtake slow vehicles in dense traffic, or fear if it engages inhabitual maneuvers. This would engender acceptability issues for the automated vehicles.

Our approach thus differs from existing methods by trying to copy human driving instead of aiming for optimal robotic driving. We also defined two objectives:Faithfully reproduce human decisions based on an artificial intelligence algorithm learning from a database. This point is important for the reasons mentioned above.Not be a purely AI-based method, which would make it a black box algorithm. When it comes to security, it is preferable to have a method whose internal behavior can be understood and any failures diagnosed.

This paper extends a previous work presented in [10]. We use the global methodology by modifying the final brick (the decision algorithm), which is fuzzy logic, by three other algorithms which have a different philosophy because they are based on learning from naturalistic driving data. We chose a decision tree, a random forest, and an artificial neural network.

This article is organized as follows: Section 2 is devoted to the state of the art of existing risk assessment and decision making for lane change maneuvers. Section 3 presents the risk assessment method and the proposed algorithm. Section 4 details the models of the decision algorithms, and Section 5 is dedicated to the discussion of the results and the comparison with the fuzzy logic algorithm. Finally, concluding remarks are addressed in Section 6.

## 2. Related Work

There are already a number of methods for assessing risk. For this task, we must estimate the future trajectories of surrounding vehicles. Many definitions of the risk can be found in the literature according to the context. In intelligent transportation systems (ITS) [11], it is usually qualified by the dangerousness of a situation for passengers who may be caused physical injury. For this, different metrics are commonly used:Inter-vehicular time [12]: Refers to the time separating two successive vehicles in the same traffic lane. The traffic law defines a two-second safety Inter-Vehicular Time (TIV). Depending on the speed, the driver must deduce their safety distance from the vehicle in front of them.“Time-To-X”: A time indication where X is a collision-related event, such as the time remaining before the impact or Time-To-Collision (TTC) [13], and it can be compared with the time required to stop the vehicle. It also can be used to warn the driver; in this case, the driver reaction time should be added to the time to stop the vehicle.An additional time indicator that is correlated with the TTC is the Time-To-React (TTR) [14], the amount of time remaining to act before the collision becomes unavoidable. In this case, the reaction time of the driver must be considered.Binary collision prediction: Future trajectories are computed for the ego-vehicle and the other vehicle. Trajectories are assumed to be calculated with sufficient accuracy (good model and exact measurements) [15].Probabilistic collision prediction: When the future motion of a vehicle is represented by a probability distribution on sample trajectories, probabilistic estimation of risks can be used by computing the collision probability between all possible pairs of trajectories; the more collision detected, the higher the risk [16]. This approach provides a lot of flexibility in the handling of uncertainties and can be adapted for any vehicle trajectory prediction model.

These methods consider that vehicles move freely in their environment without constraints and without taking into consideration traffic rules and limitations due to road infrastructure. On highways or suburban roads, for example, the variety of maneuvers is very limited compared with intersections. Moreover, these methods are objectives, and driver behavior is not considered.

For this reason, we consider that in some cases, it is more relevant to have a risk assessment method specific to one case or maneuver.

The other aspect that we deal with in this paper is a decision algorithm for lane-changing maneuvers. Rahman et al. [17] have classified lane change decision algorithms into three categories:Rule-based models: Make decisions based on physical parameters measured by the car’s sensors (radars, lidars, and cameras) such as speed and distance of surrounding vehicles, or calculated such as TTC or TTR, and rules that relate to the situation, such as the Gipps model [6], CORridor SIMulation (CORSIM) model [7], Analysis of Road Traffic and Evaluation by Micro-Simulation (ARTEMiS) model [18] and other more advanced versions [19]. The advantage of these models is that they are calculated fast in real time, but on the other hand, they can be difficult to calibrate.Artificial Intelligence: Mostly Artificial Neural Network (ANN) [8,20] and fuzzy logic [9,21]. These models are trained from databases or tuned to behave like a human driver. In addition, they can manage relatively uncertain and noisy data. Nevertheless, they have the cons of needing lots of data to be trained for ANNs and the membership functions can be difficult to define for the fuzzy logic models. Moreover, the ANN is a black box model, which means that in cases of failure, it may not be possible to pinpoint the problem and solve it.Incentive-based model: It estimates the level of benefit if the lane change is made in addition to other parameters; the model makes the decision to change lanes or not. Minimizing Overall Braking Induced by Lane change (MOBIL) [22] and Lane change Model with Relaxation and Synchronization (LMRS) [23] are the main models in this category.

In addition, there are platoon algorithms [24,25] that allow multiple vehicles in the platoon to switch lanes simultaneously.

Our final goal for this work is to obtain a hybrid decision-making algorithm that combines the ability of an artificial intelligence algorithm to learn and reproduce human behavior and overcome one of its biggest weaknesses: the fact that it is a black box model that prevents us from understanding how the algorithm works.

## 3. Methodology

The purpose of this work is to develop an incentive-based decision algorithm for lane-switching maneuvers based upon the comparison of two variables that reflect, respectively, the risk incurred when changing lanes and the benefit of switching lanes.

To this end, we use lane change samples performed by human drivers from MOOVE (MOnitoring Outillé pour le Véhicule dans son Environnement (Tooled Monitoring for the Vehicle in its Environment)) database [26], a naturalistic driving database of the Vedecom Institute that was introduced in 2015 and meant to provide a better interpretation of the environment as it would be perceived by autonomous vehicles. Over 1 million km have been gathered by using human-driven cars equipped with standard autonomous cars sensors (radars, lidar, and cameras, Figure 1). The recorded data represent all possible circulation conditions. The database includes both quantitative and qualitative variables for all the surrounding vehicles detected by the sensors and the ego-vehicles. The data are recorded in real time then imported in a data center to perform analyses and merge data from the sensors to obtain a more robust measurement.

All MOOVE vehicles are equipped with:Three technologies of exteroceptive sensors (lidar, camera, and radar).Global navigation satellite system (GNSS).Inner cameras for driver monitoring.Annotation tool for the driver.Real-time recording hardware.

To quantify the degree of risk, we take into account the longitudinal distance and relative velocity between the ego-vehicle and the vehicles in the target lane. When the distance is reduced and/or the relative speed is lowered, collision probability increases, so risk is considered higher. We use a naturalistic driving database to assess risk generally adopted by human drivers during lane change maneuvers [27]. FFor each vehicle in the future lane, we obtain a bi-variant histogram that represents the distribution of lane-changing samples in a database over longitudinal distance and relative speed to the ego-vehicle. We also split the scale into four ego-vehicle velocity ranges, 0–11 m/s, 11–22 m/s, 22–33 m/s, and 33–39 m/s, which correspond to four driving modes resulting into different behavior.

The histograms (Figure 2 and Figure 3) allow us to define a new measure for quantifying risk in a given situation: the more samples the human has performed in riskier conditions, the safer the maneuver, because it means that more drivers are prone to change lanes in these conditions. In this situation (Figure 4), and for the vehicle ahead in the target lane (vehicle 2), “riskier situation” means lower longitudinal distance and/or lower relative speed. This ends in a cumulative histogram instead of a regular histogram, characterized by Equation (Equation 1) and represented in Figure 2.
(1)$$cH_{front(low-high)}\left(i,j\right)=\sum_{n=1}^{i}\sum_{m=1}^{j}H_{front(low-high)}\left(n,m\right)$$
where $cH_{front(low-high)}$ is the matrix of the cumulative histogram for the front vehicle, and *i* and *j* are the longitudinal distance and longitudinal relative speed between the ego-vehicle and the front vehicle in the target lane, respectively. $H_{front(low-high)}$ is the bi-variant histogram. (low−high) represents the interval of the ego-vehicle’s speed corresponding to the situation.

To make the results uniform, a standardization is applied to constraint the risk between two values 0 and 1 based on the following equation:(2)$$uH_{front(low-high)}\left(i,j\right)=\frac{cH_{front(low-high)}\left(i,j\right)-min\left(cH_{front(low-high)}\right)}{max\left(cH_{front(low-high)}\right)-min\left(cH_{front(low-high)}\right)}$$
where $uH_{front(low-high)}$ is the total *n* of samples for which Interdistance<i and ΔSpeed<j (ΔSpeed is the speed difference between the ego-vehicle and the vehicle in the target lane).

Moreover, an inversion of the scale is performed, wherein 1 (which means that a maximum of drivers is inclined to make a lane change in such a situation) becomes 0 (which means minimum risk) and 0 becomes 1. The result is described by Equation (Equation 3) and represented in Figure 3.
(3)$$R_{front}\left(i,j\right)=1-uH_{front(low-high)}\left(i,j\right)$$
where $R_{front}$ is the matrix of risk for the front vehicle.

On the other side, we use a criterion we call Gain (G) that quantifies the potential gain of time if the lane change is performed and the attractiveness of the adjacent lane, and it is inspired by the work of Kestind and all [22].
(4)$$\begin{array}{c} G=\left\{\begin{array}{c} V_{ft}-V_{f}\;\;\mathrm{if}\;V_{ft}<V_{l}\\ V_{l}-V_{f}\;\;\;\mathrm{if}\;V_{ft}>V_{l} \end{array}\right. \end{array}$$
where $V_{ft}$ is the speed of the front vehicle in the target lane (vehicle 2), $V_{f}$ is the speed of the front vehicle in the same lane (vehicle 1), and $V_{l}$ is the speed limitation applied to the current road.

For the decision algorithm, the first step is to predict future trajectories of the surrounding vehicles [28,29], typically the front and rear vehicles in the target lane. For each one, the trajectory prediction is computed over 3 seconds, afterward, precision is no longer sufficient. The predicted trajectories are used to calculate the evolution of risk associated with the vehicles in the target lane (the front and the rear one if there is any), and thus obtain risk evolution curves. The risk related to the vehicle in front is expected to decline as it moves away, while the one related to the rear one is expected to rise as it moves closer. The risk associated with the whole maneuver is the maximum between the risk of the front vehicle and the rear one at each time step, and we consider the moment when the two curves intersect the optimum moment to start the maneuver and change lanes, as it is the minimum of the whole risk.

To confirm this assumption, the algorithm was tested on samples of lane-changing maneuvers extracted from the MOnitoring Outillé pour le Véhicule dans son Environnement (Tooled Monitoring for the Vehicle in its Environment) (MOOVE) database. A path prediction was made between 6 s and 3 s before lane crossing (Figure 5) to simulate the decision in real time and calculate the risk of the expected trajectory. For situations such as the one previously described, this gives us a model as described in Figure 6. The lane shift of the ego-vehicle represents the lateral distance between the center of the front bumper and the right lane marking. The discontinuity in the middle of the curve coincides with the time of the lane crossing. The top one is the evolution of risks based on the predicted trajectories over time. As we can see, the crossing moment is close to the moment when the driver starts the maneuver. In the samples we tested, it is between 0.5 and 1.5 s.

This method gives us the optimal time to initiate the maneuver, and the risk associated. However, to decide whether the lane changing should be performed or not, we need to know if the risk is worth it.

At the end, the last step is to decide if the lane change should be made or not based on the gain and the risk at the optimum instant (Figure 7). To achieve this task, we developed and compared three decision algorithms: decision tree, random forest, and artificial neural network.

Since these three algorithms are based on learning from samples, we need a database of maneuvers. We chose HighD [30] which is a dataset of naturalistic vehicle trajectories recorded on German highways using a drone. Typical limitations of established traffic data collection methods such as occlusions are overcome by the aerial perspective. Traffic was recorded at six different locations and includes more than 110,500 vehicles. Each vehicle’s trajectory, including vehicle type, size, and maneuvers, is automatically extracted. Using state-of-the-art computer vision algorithms, the positioning error is typically fewer than ten centimeters. Although the dataset was created for the safety validation of highly automated vehicles, it is also suitable for many other tasks such as the analysis of traffic patterns or the parameterization of driver models.

The extraction of positive samples (samples corresponding to lane changes) is a rather easy task. Since the lanes on which the vehicles are located are labeled on the database, it suffices to scan the entire recordings in search of the moments when the value corresponding to the number of the lane shifts.

However, for negative samples, the task is less trivial. Indeed, it is not easy to distinguish the moments during which the driver did not need to change lanes from the moments during which the driver wanted to change lanes but did not do so because of a high risk compared with the gain brought by the change in lanes. The following protocol is defined: we scan all the recordings, vehicle by vehicle, with a step of 10 seconds, and at each step we calculate the gain and the risk, if the gain exceeds the critical value, which is defined at 2, and the risk exceeds 0.5 with no lane change made within 10 seconds, we consider the sample negative. On the entire HighD database, we obtained 12.218 positive and 10.973 negative samples.

## 4. Decision Algorithms

This section is dedicated to the last brick of our decision method, which is the decision-making algorithm, which must decide whether the lane change should be made or not based on the parameters mentioned above: the risks and gain. In a previous article [10], we already used this methodology with fuzzy logic as a decision algorithm. We compare the results later.

### 4.1. Decision Tree

A decision tree is a decision algorithm representing a set of choices in a graphical form of a tree. The different possible decisions are located at the ends of the branches (the "leaves" of the tree) and are reached according to decisions made at each stage (Figure 8). A major advantage of this algorithm is that it can be automatically computed from databases by learning algorithms. These algorithms select the discriminating variables and can thus make it possible to extract logical rules which did not initially appear in the raw data.

To develop this algorithm, we used the tools implemented under Matlab, and we split the database in two (70%–30%) to perform a cross-validation. This resulted in a tree with 28 levels, and the validation on the HighD database is summarized in Table 1.

### 4.2. Random Forest

The random forest is composed of several decision trees working independently on the same problem. Each produces a decision, and it is the assembly of the decision trees that will give an overall estimate. A random forest works on the principle of bagging (Figure 9). The first step consists of dividing a dataset into subsets (decision trees), then proposing a training model to each of its groups. Finally, the results of these trees are combined in order to obtain the most robust prediction.

In our case, we developed a model with 30 learners (decision trees) and trained it on the HighD samples, and the results of the validation set are presented in Table 2.

### 4.3. Artificial Neural Network

Artificial neural networks are simple imitations of the functions of neurons in the human brain to solve machine learning problems. They can take different forms depending on the object of the data they process and according to their complexity and the data processing method (Figure 10). Furthermore, they can be used in various applications such as image processing, signal processing, language processing, control, optimization or classification.

In our case, we used an artificial neural network as a decision algorithm that takes risk and gain as input and gives a binary decision as output. We tested several architectures and parameters for the network and concluded that it is not necessary to have many hidden layers and neurons per layer because it does not improve the results. The selected network has three layers, with five neurons in the first two layers, and two in the last, and the neuron activation function is “softmax”. The obtained results are compiled in Table 3.

### 4.4. Driving Simulation

To validate the behavior of the different algorithms, we developed a driving simulator to reproduce the situations that the vehicle will face on the highway and when it will have to make the decision to change lanes or not. The graphic aspect not being a priority, the visualization is rudimentary so as not to weigh down the simulation (Figure 11).

The simulation is divided into two parts, the first is a scenario generator, which sets the initial conditions of the simulation. The modified variables are the longitudinal position of the 4 vehicles (and thus the inter-distances), the speeds of each vehicle, the speed limit applied to the road, and whether the two vehicles on the target lane exist.

The second part is the simulator itself, which calculates the trajectories of the four vehicles and calculates the decision-making algorithm in parallel. If the latter decides to change lanes, then the ego-vehicle performs the maneuver and the simulator checks that there is no collision. Each scenario is then saved for post-processing (Figure 12). A total of 3740 scenarios was generated and for each scenario the four algorithms were tested. Table 4 represents the rate of positive and negative decisions by the different algorithms on all the scenarios generated.

In order to obtain a representation of the risk taken by the algorithms, we drew Figure 13 which represents the histogram of the percentage of maneuvers carried out with a risk higher than the level of the bar for all the algorithms.

## 5. Discussion

In the light of all the results compiled in the previous section, we noticed that the algorithms developed in this paper exceed the results obtained with fuzzy logic algorithms when the evaluation criterion is the fidelity of the decisions compared with the database HighD, and therefore fidelity in relation to human behavior. This is normal and expected knowing that these algorithms learned from samples from the same database that is used to evaluate them, with the best result obtained with random forest. It is difficult, according to us, to obtain better result on this evaluation criterion for several reasons. First, because the different used samples did not present the same driver, which leads to behavioral differences. Moreover, the driving conditions are not taken into account; the behavior may be different at night or in rainy weather, and finally, there may be some drivers in the database who behaved very dangerously at certain times. However, this is not statistically significant and is drowned in the other samples with a more classic behavior, and is therefore not learned by the models developed.

For the obtained results with the driving simulator, two interesting remarks are noted: First, the table shows that the algorithms behave very similarly, with close positive and negative decision rates, with a noticeable difference in fuzzy logic. This is again explained by the fact that the first three algorithms are based on learning, while the last one is based on manual adjustment. Secondly, the table tells us about the similarity of behavior between the algorithms but does not tell us about the level of risk taken during the maneuvers. To this end, we must look at the histograms in Figure 13. We notice the same trend for all the algorithms but also that fuzzy logic takes more risk than the other algorithms, which shows that the algorithms developed in this article are not only more similar to human behavior but also that they take less risk within the simulator. We can also notice that the lane change rate for a risk greater or equal to 0.95 is lower than 0.67% for all the algorithms, with a minimum established at 0.45% for random forest. We therefore conclude that random forest is the one that obtained the best results among those compared in this paper.

## 6. Conclusions

This paper presents a continuation of a methodology for risk assessment and decision making for lane-changing maneuvers. The first part is to assess the risk associated with the maneuver. For each vehicle in the target lane (the front and rear vehicle), we performed a trajectory prediction over 3 seconds to determine the best moment to start the maneuver. Secondly, the decision algorithm is used to decide whether or not the decision should be performed. We chose artificial neural networks, decision trees, and random forests, which we also compared with fuzzy logic. These different algorithms learned from samples from the HighD database.

It appears that the three algorithms obtain better results than fuzzy logic on the database. They were also tested in a developed driving simulator, which showed that the different algorithms behave in an almost similar way. There is also an improvement in risk taking during maneuvers; in fact, the difference is notable for the benefit of the three algorithms developed. Future work will consist of experimental validation of the results obtained with Vedecom vehicles.

## Figures and Tables

**Figure 1 sensors-22-08148-f001:**
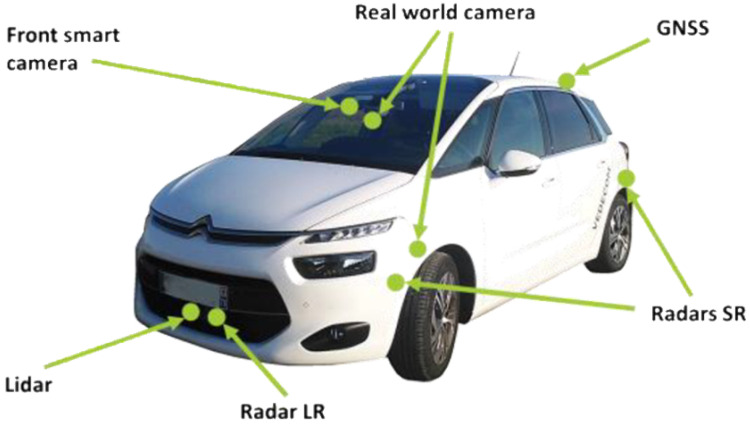
MOOVE vehicle.

**Figure 2 sensors-22-08148-f002:**
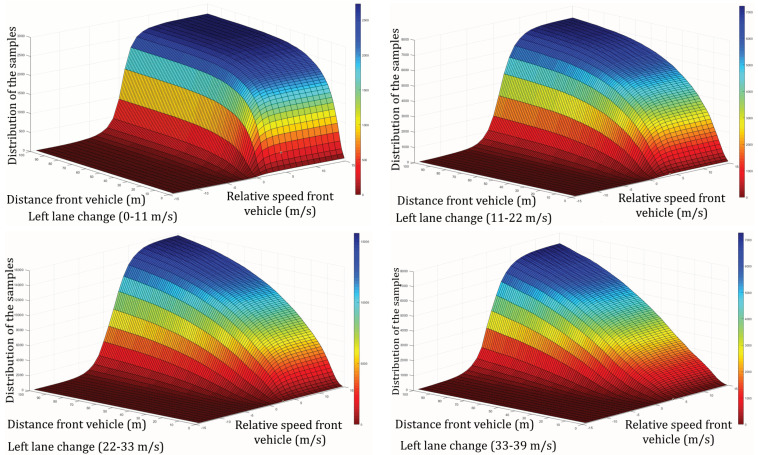
Bi-variant cumulative histogram divided in 4 speed intervals for the front vehicle.

**Figure 3 sensors-22-08148-f003:**
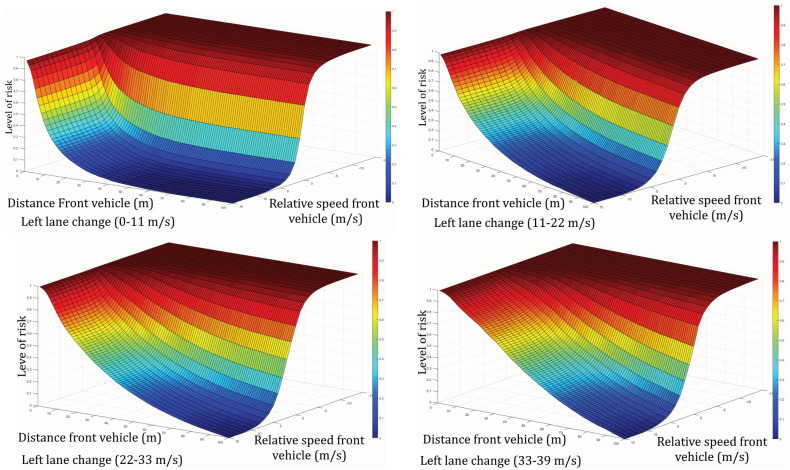
Standardized bi-variant cumulative histogram divided in 4 speed intervals for the front vehicle.

**Figure 4 sensors-22-08148-f004:**
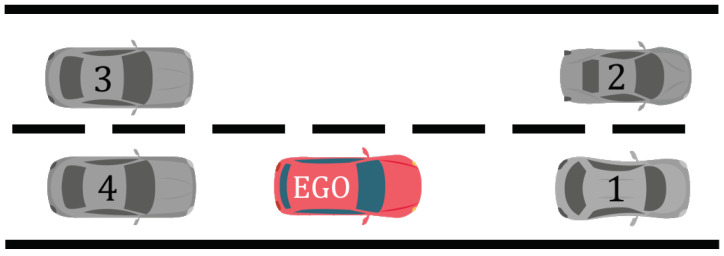
Road configuration with the surrounding vehicles before the maneuver (red: ego-vehicle).

**Figure 5 sensors-22-08148-f005:**
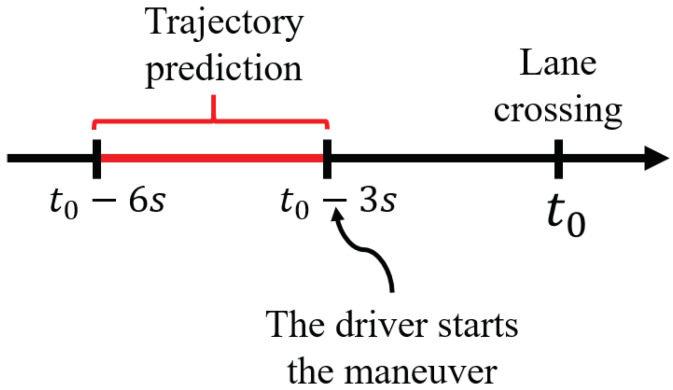
Timeline of the predicted trajectories.

**Figure 6 sensors-22-08148-f006:**
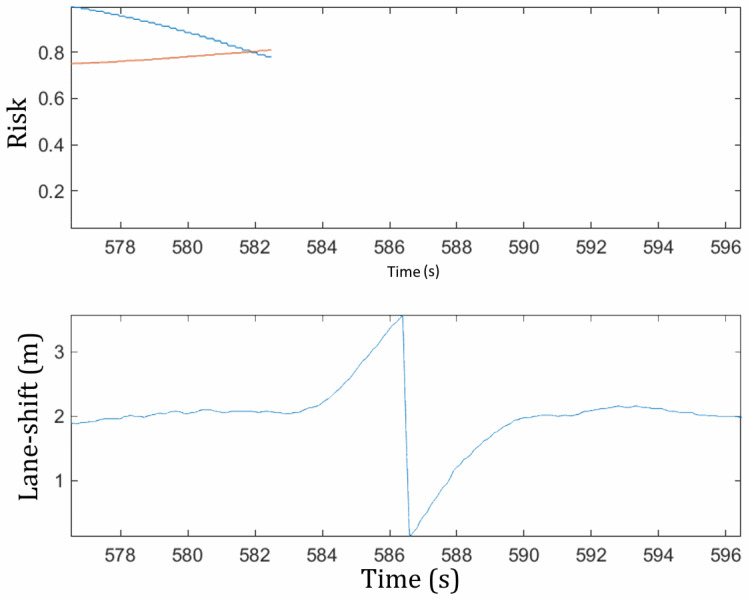
Evolution of the risk for the front vehicle (blue), the rear vehicle (red), and the lane shift of the ego-vehicle.

**Figure 7 sensors-22-08148-f007:**
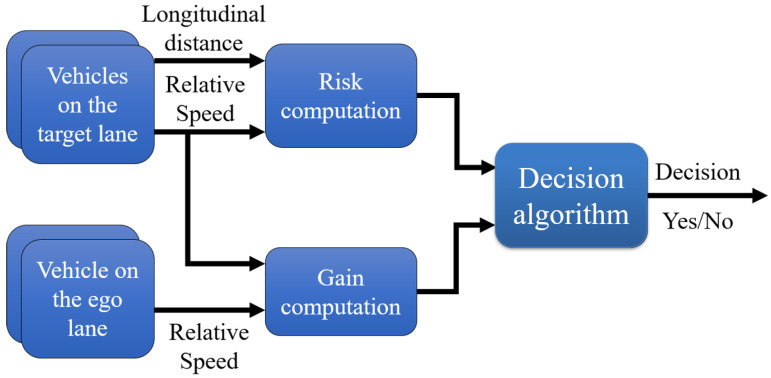
Diagram of the decision algorithm.

**Figure 8 sensors-22-08148-f008:**
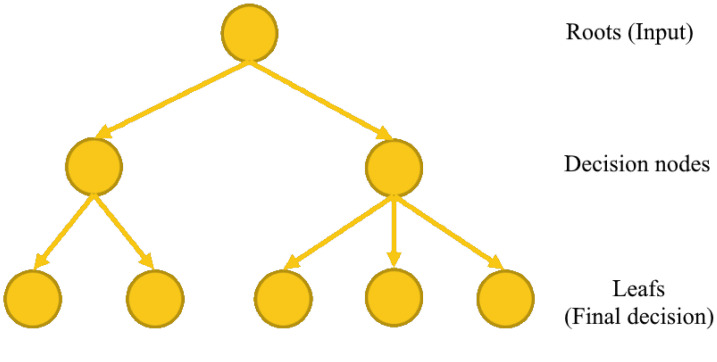
Decision tree structure.

**Figure 9 sensors-22-08148-f009:**
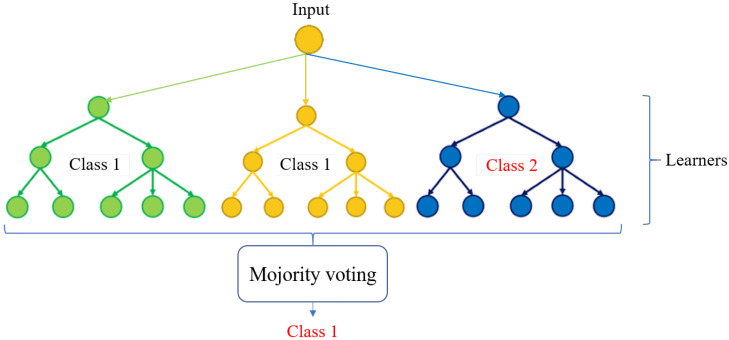
Random forest structure.

**Figure 10 sensors-22-08148-f010:**
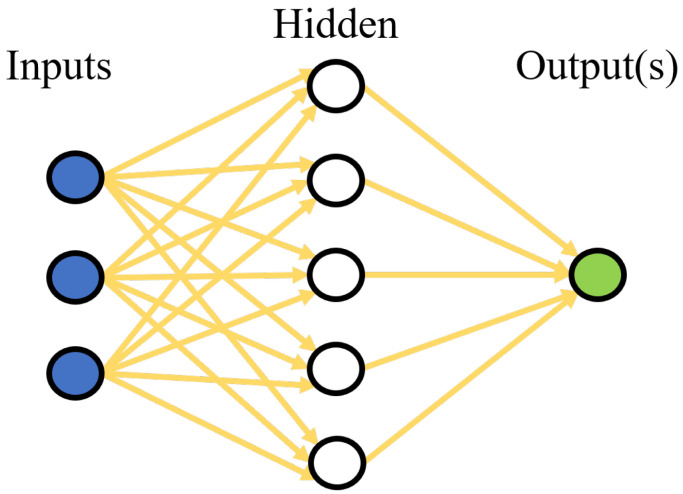
Artificial neural network structure.

**Figure 11 sensors-22-08148-f011:**
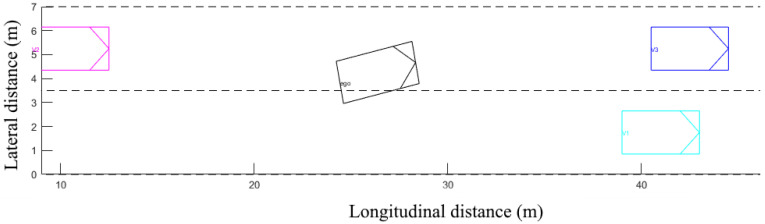
Graphical representation of the driving simulator.

**Figure 12 sensors-22-08148-f012:**
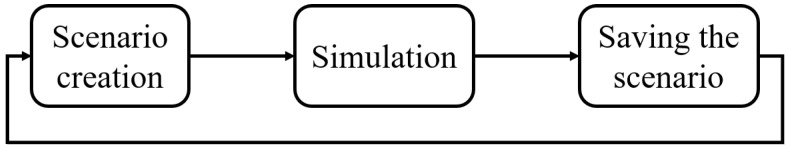
Diagram of driving simulator.

**Figure 13 sensors-22-08148-f013:**
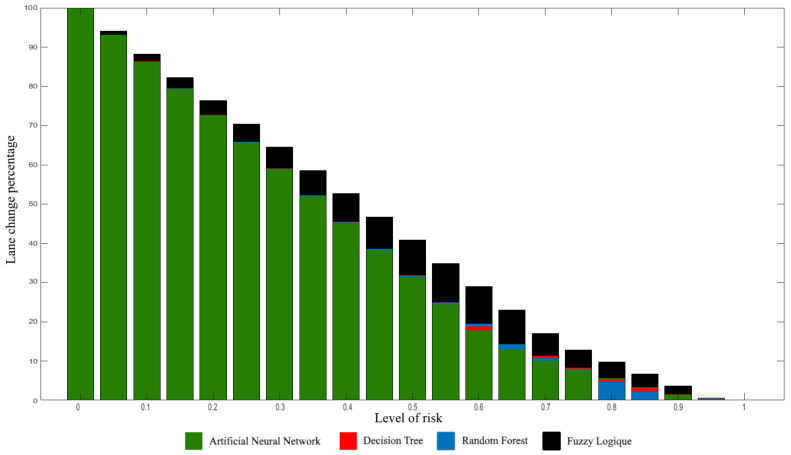
Histogram of the level of risk taken by each algorithm.

**Table 1 sensors-22-08148-t001:** Confusion matrix for decision tree.

		Decision Tree	
		Positive	Negative
**Real Decision**	Positive	**75.53%**	24.47%
	Negative	21.64%	**78.36%**

**Table 2 sensors-22-08148-t002:** Confusion matrix for random forest.

		Random Forest	
		Positive	Negative
**Real Decision**	Positive	**87.3%**	12.7%
	Negative	20.7%	**79.3%**

**Table 3 sensors-22-08148-t003:** Confusion matrix for artificial neural network.

		ANN	
		Positive	Negative
**Real Decision**	Positive	**75.84%**	24.16%
	Negative	21.50%	**78.50%**

**Table 4 sensors-22-08148-t004:** Positive and negative decision rate for each algorithm in the simulator.

	Positive	Negative
Decision Tree	37.06%	62.93%
Random Forest	36.07%	63.93%
ANN	36.58%	63.42%
Fuzzy Logic	33.11%	66.89%

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
