# Peer review of "Detection and Risk Analysis with Lane-Changing Decision Algorithms for Autonomous Vehicles"

_sensors, 2022, doi:10.3390/s22218148_

Round 1

Reviewer 1 Report

Abstract:

The research problem is not clear. What triggered the authors to decide to improve the decision making algorithm? (because autonomous driving still faces many challenges). What is the problem with exiting decision making algorithms that we need to improve one. Abstract Section should contain enough information to outline the scope of paper to the readers. 

what is the aim? to improve an existing algorithm, or ...

Introduction:

line 29: "Nevertheless, they are not widely available". 

What do you mean by "they"? Do you mean different levels? Is not Level 0 and 1 widely available?

Lines 34, 35: "Improving the perception of the environment to provide a correct interpretation of the information returned by the sensors whatever the lighting and weather conditions".

Not easy to understand this statement. 

Again I was expecting to read more about the specific research problem/question in the Introduction section, what motivated the authors to work on this project?

The research problem is very broad as "autonomous driving still faces many challenges", then we have decided to improve decision-making algorithm.

Please let me know what I am missing.

Author Response

Dear M.

For this paper, we had two objectives. the first was to develop a risk assessment metric associated with lane change maneuvers. We were able to develop a different approach thanks to the MOOVE database which offers us a large number of maneuvers, which we do not find in any other database. This latter is differentiated by the fact that it is based on real maneuvers performed by humans and does not subsequently need to be calibrated or attached to thresholds to define what is considered dangerous. 1 being the absolute risk of collision, and 0 the total absence of risk.

Q1:What do you mean by "they"? Do you mean different levels? Is not Level 0 and 1 widely available?
By "they" we mean the fully autonomous vehicles, we made a change to the sentence to make it clearer.

Q2: Lines 34, 35: "Improving the perception of the environment to provide a correct interpretation of the information returned by the sensors whatever the lighting and weather conditions".
Not easy to understand this statement.

By this, we mean that the perception algorithms that analyze the information coming from the sensors still need to be improved, because in many accidents that take place while the vehicle was in autonomous mode, occur due to poor recognition of objects (vehicles, trucks, road curvature ...) 

Reviewer 2 Report

This article developed a three-step decision-making method, namely trajectory prediction, risk and gain computation, and finally decision-making for lane changing on highways and sub-urban roads. The topic is hot and very interesting. There are a few comments for your references:  

1, The introduction could be improved to erase common sense ( L1-L5 level).

2, The related work could be further explained by introducing related references. 

3, The quality of the figures could be improved. 

4, The fundamental of the RF and ANN could be introduced in brief. 

5, Minor spell check are needed in Conclusion and other sections. 

Author Response

Dear Mr.

Q1: The introduction could be improved to erase common sense ( L1-L5 level).
We think it is relevant to define the different levels of autonomy defined by the SAE.

Q2. The related work could be further explained by introducing related references.
We have improved the state of the art by incorporating more recent references.

Q5. 5, Minor spell check are needed in Conclusion and other sections. 
We have corrected a large number of language errors.

Round 2

Reviewer 1 Report

Unfortunately, I do not see any changes in the abstract section and not enough changes in the introduction section to properly articulate the research problem and the motivation for your work.

The research problem is not clear. The focus of the work is mainly on the methodology. But for what reason, I could not find it. What triggered the authors to decide to improve the decision-making algorithm? (Because autonomous driving still faces many challenges). What is the problem with exiting decision-making algorithms that we need to improv.. Abstract Section should contain enough information to outline the scope of paper to the readers.

In which lines I can find your responses to these questions?

Author Response

Please see the attachment:

Round 3

Reviewer 1 Report

I can see you have done a good job to develop your algorithm; it needs to be well presented as well. Please try to follow academic-style writing, it means that readers could easily follow and read your work by understanding all steps you have taken.

I suggest you to have a look at papers written in social science e.g. psychology; you will find how easy and clear the structure of papers are even for general readers.